# Could Targeted Pharmacotherapies Exert a “Disease Modification Effect” in Patients with Chronic Plaque Psoriasis?

**DOI:** 10.3390/ijms232112849

**Published:** 2022-10-25

**Authors:** Francesco Bellinato, Andrea Chiricozzi, Stefano Piaserico, Giovanni Targher, Paolo Gisondi

**Affiliations:** 1Section of Dermatology and Venereology, Department of Medicine, University of Verona, 37129 Verona, Italy; 2UOC di Dermatologia, Dipartimento di Scienze Mediche e Chirurgiche, Fondazione Policlinico Universitario A. Gemelli IRCCS, 00144 Rome, Italy; 3Dermatologia, Dipartimento di Medicina e Chirurgia Traslazionale, Università Cattolica Del Sacro Cuore, 00168 Rome, Italy; 4Section of Dermatology, Department of Medicine, University of Padua, 35122 Padua, Italy; 5Section of Endocrinology, Diabetes and Metabolism, Department of Medicine, University of Verona, 37129 Verona, Italy

**Keywords:** psoriasis, quality of life, psoriatic arthritis, resident memory T cell, biologics

## Abstract

Chronic plaque psoriasis is an immune-mediated skin disease with a chronic relapsing course, affecting up to ~2–3% of the general adult population worldwide. The interleukin (IL)-23/Th17 axis plays a key role in the pathogenesis of this skin disease and may represent a critical target for new targeted pharmacotherapies. Cutaneous lesions tend to recur in the same body areas, likely because of the reactivation of tissue-resident memory T cells. The spillover of different pro-inflammatory cytokines into systemic circulation can promote the onset of different comorbidities, including psoriatic arthritis. New targeted pharmacotherapies may lead to almost complete skin clearance and significant improvements in the patient’s quality of life. Accumulating evidence supports the notion that early intervention with targeted pharmacotherapies could beneficially affect the clinical course of psoriatic disease at three different levels: (1) influencing the immune cells infiltrating the skin and gene expression, (2) the prevention of psoriasis-related comorbidities, especially psoriatic arthritis, and (3) the improvement of the patient’s quality of life and reduction of cumulative life course impairment. The main aim of this narrative review is to summarize the effects that new targeted pharmacotherapies for psoriasis may have on the immune scar, both at the molecular and cellular level, on psoriatic arthritis and on the patient’s quality of life.

## 1. Introduction

Plaque psoriasis is a chronic inflammatory skin disease that, similarly to other immune-mediated inflammatory diseases, is associated with a systemic inflammatory burden and various comorbid conditions, including psoriatic arthritis (PsA) [1]. Psoriatic skin lesions are characterized by erythematous and scaly plaques usually occurring on the scalp and extensor surfaces of upper and lower limbs (Figure 1) and a tendency to recur at the same body sites [2]. Older anti-psoriatic therapies are usually effective in resolving disease manifestations but may negatively affect the patient’s quality of life. Though these therapies are able to clear psoriasis skin lesions, a residual “immune scar” is detected in resolved lesional skin and may predispose to the recurrence of psoriatic lesions [3,4,5,6,7,8]. Conversely, new targeted pharmacotherapies for psoriasis may lead to the clearance of skin lesions and a reduction in the immune scar at the molecular level. Preliminary evidence supports the concept of a differential impact of each targeted pharmacotherapy on both disease activity and immune scarring [4,9]. This narrative review is aimed at summarizing the effects of new targeted pharmacotherapies for psoriasis on the immune scar (both at the molecular and cellular level) and the tissue-resident memory T cells in the pathogenesis of comorbid conditions, as well as the effects of these new targeted pharmacotherapies on both psoriasis comorbidities and patient’s quality of life (Table 1).

## 2. Immunopathogenesis of Psoriasis

The complex pathogenic mechanisms underlying the development of psoriatic plaques are typically characterized by the activation of both adaptive and innate immunity [21]. Different subsets of T cells, dendritic cells, mast cells, neutrophils, and innate lymphoid cells-3 may promote the massive production of both chemokines and cytokines, mostly belonging to the interleukin (IL)-23/IL-17 axis [21]. In particular, these two cytokines are highly expressed in patients with psoriasis and are recognized to be pathogenically and therapeutically relevant. IL-23 is a regulatory cytokine acting on the T cell compartment, stimulating the expression of RAR-related orphan receptor γ (RORγ) and the production of IL-17A and other effector cytokines, such as IL-17F and IL-22. IL-17A is the main effector cytokine that induces skin inflammation by acting on tissue cells, conversely to IL-23, which acts on immune cells bearing the IL-23 receptor [22,23,24]. IL-17A is expressed by neutrophils and mast cells, though the centrality of T cell subsets has been widely demonstrated [25,26]. Activated CD8+ T cells are crucial in the early phases of psoriatic inflammation, as demonstrated by immunosuppressed mouse models xenografted with psoriatic non-lesional skin that developed psoriatic lesions after the expansion and migration of CD8+ T cells within the epidermis [27,28]. These oligoclonal T cells located within the epidermis consist of effector cells producing multiple pro-inflammatory and pro-proliferative cytokines that could also react against keratinocyte- or melanocyte-derived self-antigens [29,30,31]. Beside CD8+ T cells, another pathogenically relevant source of both IL-17A and IL-22 is represented by CD4+ T cells (T helper [h] 17 cells) [24,32,33]. The differentiation and activation of IL-17-producing T cells are mostly driven by IL-23, which is composed of the p19 and p40 subunits [34]. The presence of IL-23 is required for the development of psoriatic lesions in various preclinical models [35,36]. IL-23 is mainly produced by myeloid dendritic cells (DCs) and Langerhans cells, whose activation occurs through toll-like receptor (TLR)-mediated signals, and by IFN-α, a type I interferon, produced by plasmacytoid DCs [37,38]. This latter DC subset might react against self-nucleic acids bound to antimicrobial peptides or be stimulated by IL-36-binding TLR-9 [38,39]. The overall production of IL-23 is thought to expand and maintain T17 cells, including those belonging to the tissue-resident memory T cell (T_RM_) compartment.

### 2.1. Immune Memory Characterizing Psoriatic Skin

T_RM_ cells constitute a large, long-lived, and resident T cell subset with low migration activity bearing CD49a, which constitutes the α-subunit of the α1β1 integrin receptor (also known as very late antigen 1, VLA-1), and other cell surface markers, such as CD69 and CD103 (i.e., a ligand for E-cadherin, which is expressed by epidermal keratinocytes) [40]. Increasing evidence suggests a central role of T_RM_ cells in the recurrence of psoriasis [5,40,41,42]. Skin T_RM_ cells embedded in a human skin graft undergo local proliferation and infiltration in the dermis and epidermis of hosting immunodeficient mouse models, determining the onset of psoriasis lesions. When psoriatic non-lesional human skin is grafted onto an immunodeficient mouse model, the occurrence of psoriasis lesions is associated with the expansion of human CD8+ T_RM_ cells [28]. Psoriatic lesional and non-lesional skin is infiltrated by a greater number of CD8+ T_RM_ cells compared with normal skin, and the increased number of infiltrating CD8+IL-17A+ T_RM_ cells, compared with IFN-γ-producing cells, correlates with disease duration [28,41,43]. Apparently-healed skin contains CD8+ T_RM_ cells that are mostly localized at the epidermis (as they express CD103), while CD4+ T_RM_ cells localize in the dermis [44,45].

Moreover, a large pool of α/β T_RM_ cells expressing IL-17A alone, IL-17A and IL-22, or IL-22 alone persist in post-lesional psoriasis skin [5]. Cytokine production is not exclusively oriented toward a T-17 phenotype as T_RM_ also express IFN-γ, showing a T1 phenotype that could be induced by IL-15 stimulation [46]. Nevertheless, additional findings clarified that cytokine production might be specifically determined by the inflammatory condition affecting the skin. Indeed, activated T cells in skin explants showed an upregulation of IFN-γ signaling, regardless of the inflammatory disease, whereas IL-17 upregulation is preferentially observed in psoriatic skin [10].

### 2.2. Molecular Scarring in Post-Lesional Psoriatic Skin

The first transcriptomic study investigating the molecular scar in etanercept-treated resolved psoriatic skin identified a cluster of 248 genes that were not normalized by the drug [3]. This cluster included IL-12, IL-22, IL-17, IFN-γ, and IFN γ-signature genes (i.e., Mx1 encoding interferon-induced GTP-binding protein Mx1). Other transcriptomic studies revealed a greater ability of the treatment with IL-17 antagonists to reduce the molecular scar, showing a larger set of normalized genes compared to treatment with either anti-TNF agents or anti-p40 IL-12/IL-23 agents [4,47]. Ixekizumab normalized a ~6-fold larger set of genes compared to etanercept, whereas brodalumab in responder patients obtained normalization of ~90% of altered genes associated with the activation of neutrophils, DCs, and Th17 cells, with a greater magnitude compared to ustekinumab and etanercept [4,47]. The molecular scar suggested a residual, latent activation of immune cells in the resolved skin, even though each therapeutic agent seemed to have a different ability to reduce the molecular scar and also interfere with immune memory T cells.

## 3. Impact of Targeted Pharmacotherapies on the Tissue-Resident Memory Compartment

Preliminary data analyzing the effect of anti-psoriatic therapies on immune skin memory were obtained by patients treated with narrow-band UVB, ustekinumab, or infliximab [6]. A subset of epidermal CD8+ T cells expressing CCR6 (binding to CCL20 and mostly expressed by IL-17A-producing cells), CD103, and IL-23R was enriched in resolved psoriasis lesions [6]. Those cells expressing CD103 responded to ex vivo stimulation with IL-17 production, and this subset of T cells were more frequently detected after narrow-band UVB compared to biological medications, suggesting that CD49a-expressing epidermal CD8+ cells represent a stable, resting population that retain the ability to produce IL-17, while IL-22 would be mainly produced by skin-residing CD4+ T cells [6]. The more recently introduced classes of biologic agents targeting either IL-17 or IL-23 increased the threshold of optimal therapeutic response, demonstrating higher rates of response and superiority compared to previous biologic agents. IL-23 inhibitors are considered potentially disease-modifying drugs more than IL-17 antagonists because of preliminary pathogenic, clinical, and molecular findings [11,48,49]. IL-23 is a regulatory cytokine driving the development, maintenance, and activation of IL-17-producing T cells. IL-23 inhibition is thought to modulate the T cell compartment, causing a reset of pathogenic inflammatory T cells and an increase in the rate of non-inflammatory T regulatory (T_reg_) cells. Because IL-17 is an effector cytokine acting on tissue cells, its inhibition causes indirect effects on the T cell compartment with a lower modulatory ability compared to IL-23 inhibitors, although it lessens IL-23 expression.

Clinical trials testing IL-23 inhibitors showed long-lasting maintenance of the therapeutic response following treatment discontinuation compared to IL-17 inhibitors [50,51,52,53]. In particular, clinical remission was maintained for over 24–30 weeks after IL-23 inhibitor treatment withdrawal, beyond their elimination half-lives, in a high proportion of patients; with IL-17 blockers (i.e., ixekizumab and bimekizumab), this effect was seen for a shorter period of time (approximately 20 weeks) [48]. A head-to-head clinical trial comparing guselkumab and secukinumab provided relevant insights about the different effects of these two drugs on the T_RM_ compartment and, more in general, on skin immune memory [11]. In fact, after a 24-week treatment with guselkumab or secukinumab, the number of CD4+ and CD8+ T_RM_ cells decreased in psoriatic lesions of both treatment arms; in contrast, guselkumab reduced memory T cells while maintaining T_reg_ cells and vice versa for secukinumab treatment [11]. Secukinumab decreased the number of T_reg_ cells in a more pronounced way than guselkumab [11]. In addition, a significantly greater decrease in Langerhans cells (a source of IL-23 in lesional psoriatic skin) infiltrating post-lesional skin was observed in guselkumab-treated versus secukinumab-treated patients. No other significant differences in terms of DC and macrophage modulation were found [11]. These findings suggest that the increased T_reg_/CD8+ T_RM_ ratio may be related to the superior long-term control of skin inflammation achieved by inhibiting IL-23. Because the infiltration of CD8+ T_RM_ correlates with the disease duration, the therapeutic intervention of neutralizing IL-23 should be timely and as early as possible. This hypothesis is tested in the GUIDE study where patients with short disease duration (<2 years) were treated with guselkumab, verifying whether these patients could maintain longer drug-free control of the disease after treatment discontinuation and whether the patients who demonstrated a fast and strong initial response to guselkumab could maintain disease control with less frequent dosing (every 16 weeks instead of every 8 weeks) [54]. The differential effects of guselkumab and secukinumab on the T_RM_ compartment are reported in Figure 2.

## 4. T_RM_ Compartment in the Pathophysiology of Psoriatic Arthritis

The pathogenic aspects suggesting the progression of psoriasis (skin inflammation) to psoriatic arthritis (i.e., enthesis and synovial inflammation) are limited but include the T_RM_ cell compartment. This process may be driven by pathogenic cytokines such as IL-23. Transgenic mice overexpressing IL-23 in the skin developed psoriasis-like skin (histologically characterized by acanthosis, parakeratosis, hyperkeratosis, and inflammatory infiltrates in the dermis) that preceded the development of the articular manifestations [55]. Production of IL-23 by keratinocytes induced an inflammatory process at the joint and enthesis, likely through the pro-inflammatory effect of IL-17, since IL-22 (a cytokine downstream of the IL-23 pathway increased in the serum of such transgenic mice) proved to have a protective role against the development of PsA [55]. Indeed, IL-22 deficiency did not affect skin disease development, but it aggravated PsA development [55]. Further studies are needed to better define the contribution of pathogenic mediators in inducing a different trafficking activity toward the joint synovial tissue and enthesis of circulating CD8+ T_RM_ cells expressing CCR4 and CXCR3, which were found to be increased in the peripheral blood of psoriasis patients compared to healthy controls [56]. Other studies revealed an increased number of circulating memory CD8+ T cells amongst psoriatic patients with PsA compared with both healthy controls and patients with psoriasis alone [57]. Abundant CD8+ cells were also found in the synovial fluid of PsA patients, showing a more marked clonal expansion compared to CD4+ T cells [58,59]; these expanded CD8+ T cells found in the joint could be considered memory cells expressing tissue-homing and tissue-resident markers on the basis of transcriptomic findings [59]. Previous findings identified the overexpression, both in the skin and synovium, of antimicrobial peptides, in particular LL37, as pathogenic triggers of the activation of similar T cell clones in both tissues of PsA patients. LL37 was also targeted by antibodies that were detected in both the synovial fluid and plasma of PsA patients, though the precise role of such autoantibodies in the pathogenesis of PsA needs further investigation.

## 5. Targeted Pharmacotherapies May Prevent the Transition from Psoriasis to PsA

### 5.1. Transition from Psoriasis to PsA

Chronic plaque psoriasis precedes the inception of PsA in most patients by an average time span of nearly 7–10 years, representing one of the strongest risk factors for PsA [60,61]. It has been suggested that the onset of joint disease derives from the interaction between different environmental factors in genetically susceptible individuals evolving through three clinically quiescent phases [61,62]. Initially, abnormal activation of the immune system originating from the skin or the intestinal mucosa extends to the entheses and joints (i.e., the preclinical phase) [62]. Then, imaging findings can be detected by ultrasonography and/or magnetic resonance in the absence of clinical symptoms (the subclinical phase). Indeed, ultrasonographic enthesopathy was found to be common among patients with psoriasis without any clinical signs of arthritis. As an example, tendon thickness and the Glasgow Ultrasound Enthesitis Scoring System (GUESS) score evaluated at common entheseal sites were found to be significantly higher in psoriatic patients without musculoskeletal complaints than in healthy controls [63,64]. Finally, non-specific symptoms, such as arthralgia and fatigue, can be reported without any evidence of synovitis or enthesitis during the physical examination (the prodromal phase) [62]. A higher incidence of sonographically detected tenosynovitis was found in psoriasis patients with arthralgia compared to their counterparts without arthralgia [65]. Additionally, sonographical signs of enthesitis are able to predict the development of PsA [65].

To date, different genetic and modifiable risk factors for PsA transition have been identified, although some controversial findings have also been observed. These include the HLA and non-HLA genetic variants involved in the immune response, disease severity, specific site involvement (scalp, nails, folds), mechanical factors triggering an exaggerated inflammatory response (so-called mechano-inflammation), obesity, dyslipidemia, and intestinal dysbiosis [66]. Recently, Ogdie et al. validated a model for predicting the risk of developing PsA in patients with psoriasis. The optimal model included six variables (namely, the Psoriasis Epidemiology Screening Tool (PEST), body mass index, modified Rheumatic Disease Comorbidity Index, work status, alcohol consumption, and patient-reported fatigue) and predicted the development of PsA within 24 months with a sensitivity of 82.9% and a specificity of 48.8% [67].

Psoriasis and PsA share various pathophysiological similarities in terms of tissue microanatomy and immunological mechanisms. Both skin and joints contain avascular sites, namely the epidermis and fibrocartilage zone, that are subject to the Koebner phenomenon [68]. In treating both psoriasis and PsA, the systemic drugs that have shown to be effective include conventional disease-modifying antirheumatic drugs(DMARDs) such as methotrexate, targeted small molecules such as PDE4 inhibitors, and different biological targeted therapies [69]. TNF-α is involved in several phases of the pathogenesis of psoriasis and PsA and has been the first successful target of cytokine-mediated therapy, as reported in large, randomized, controlled clinical trials [70]. Innate and adaptative cells sustaining the IL-23/17 cytokine axis are pathogenically crucial in both conditions and are relevant targets of anti-IL-17 and IL-23 biological agents. These observations support the concept that treatment with targeted pharmacotherapies might be able to induce skin clearance of psoriatic lesions and reduce the risk of PsA development [71]. Conversely, skin-directed therapies such as topical agents and narrow-band UVB phototherapy seem to have a beneficial impact on the cutaneous disease but do not appear to act systemically in preventing PsA development.

### 5.2. Reducing the Risk of Developing PsA in Patients with Psoriasis

Promising results from some early studies suggest that the ambitious goal of modifying the clinical course of psoriasis may be achievable through targeted pharmacotherapies [71]. In fact, observational studies showed that systemic treatment of moderate-to-severe psoriasis might reduce the risk of developing PsA. Savage et al. provided the first evidence that biologics are effective on subclinical enthesopathy and may also have the potential to prevent PsA development [12]. The impact of systemic treatment with ustekinumab on sonographic features of subclinical enthesopathy was investigated in a 52-week open-label study of 23 PsA patients. The mean inflammation scores decreased significantly by 42% at week 24 and by 47.5% at week 42 [12]. Kampylafka et al. investigated whether IL-17A inhibition in psoriatic patients with subclinical inflammatory joint changes may interrupt the progression from psoriasis to PsA [13]. In the Interception in Very Early PsA (IVEPSA) study, a single-arm prospective open-label study, the authors examined the effect of secukinumab on the inflammatory and structural changes of peripheral joints in psoriatic patients with arthralgia (without established PsA). Of the 20 patients included in the study, treatment with secukinumab for 24 weeks was associated with significant improvements in arthralgia, psoriatic arthritis magnetic resonance imaging scoring system, and synovitis subscores, while erosions and enthesophytes did not change [13]. Further evidence derives from some real-life observational studies that assessed the risk of new-onset PsA in cohorts of patients receiving targeted pharmacotherapies versus skin-directed therapies (phototherapy and/or topical therapy).

Gisondi et al. assessed the incidence of PsA in a cohort of 464 patients with psoriasis who received continuous treatment with biological disease-modifying antirheumatic drugs (bDMARDs) compared to phototherapy treatment [14]. A significantly lower annual incidence rate of PsA was found in patients receiving bDMARDs versus those treated with phototherapy, 1.20 (95% CI 0.77 to 1.89) versus 2.17 (95% CI 1.53 to 3.06) cases per 100 patient-years, respectively. Treatment with bDMARDs was associated with a lower risk of PsA incidence (adjusted hazard ratio 0.27, 95% CI 0.11–0.66) [12] (Figure 3).

Acosta Felquer et al. compared the incidence of PsA in a cohort of 1719 patients with psoriasis receiving topics/no treatment, conventional disease-modifying antirheumatic drugs or biologics [15]. Interestingly, the risk of incident PsA in patients with psoriasis treated with biologics was significantly lower (risk reduction = 0.26; 95% CI 0.03–0.94; *p* = 0.011) compared to those treated only with topicals but not in comparison to those treated with conventional DMARDs (risk reduction = 0.35; 95% CI 0.035–1.96; *p* = 0.1007). Adjusted Cox proportional hazards regression analysis showed that biologics use was a protective factor for PsA development (hazard ratio = 0.19; 95% CI 0.05–0.81) [15]. Solmaz et al. performed a retrospective chart review on 203 psoriasis patients with musculoskeletal symptoms who were referred for rheumatological assessment [16]. Patients receiving target therapies showed lower rates of PsA onset (12% for biologics and 9.6% for cDMARDs) compared to those treated with topicals or non-treated cases (37.4%, *p* < 0.001). Moreover, among patients diagnosed with PsA, none of the patients treated with biologics had dactylitis, as opposed to 28.6% of those treated with cDMARDs and 48.6% of those who were under no or only topical treatments (*p* = 0.046) [16]. These findings suggest that new symptoms and signs leading to PsA diagnosis seem to decrease with the use of systemic treatments. In a nested case-control study by Rosenthal et al., a total of 663 psoriatic patients not diagnosed with PsA who were treated with biologics were compared to as many controls not receiving biologic treatments. The control group showed an increased risk for PsA compared to patients under biologic treatments within 10 years of follow-up (adjusted HR 1.39, 95% CI 1.03–1.87) [17].

At first glance, the results of a study by Meer et al. examining the association of biologic therapy use for psoriasis with the risk of incident PsA seem to differ from previous findings [72]. According to this retrospective cohort study based on an electronic database involving 193,709 patients with psoriasis without PsA and initiating therapy for psoriasis (oral, biologic, or phototherapy), the incidence of PsA was 77.3 per 1000 person-years among those treated with biologics, 61.9 among patients treated with oral therapy, 26.1 among those on phototherapy and 5.8 per 1000 person-years among patients without a prescription for one of the target therapies. Adjusted HR for biologic users was 4.48 (95% CI 4.2 to 4.7) compared with oral or phototherapy users [72].

Several biases should be considered when investigating the association between exposure to targeted therapies and the risk of PsA in patients with psoriasis. Firstly, confounding by indication should be considered, i.e., when the choice of a therapy for a given patient is motivated by many factors, including psoriasis severity, but also comorbidities, prior therapy, and potential unmeasured confounders [73]. Protopathic bias should also be considered, i.e., when a pharmaceutical agent is inadvertently prescribed for early manifestations of a disease that has not yet been detected, including PsA. The accuracy of PsA diagnosis may also be difficult to confirm (ascertainment bias) because of the heterogeneity of the disease presentation and the lack of specific biomarkers. Finally, the survival bias should be considered, as patients without PsA persist in receiving biologic therapy, being closer to PsA development, increasing the observed risk of PsA in the biologic group [72]. Patients need to be followed up longitudinally for a long period because arthritis develops on average several years after the initial diagnosis of psoriasis [63]. Planning a randomized trial is scarcely feasible and unethical because it would imply enrolling two groups of patients with moderate-to-severe psoriasis and comparing the effect of treatment versus no treatment for a very long time. Even if the practicability of randomized clinical trials is undermined by several issues, the risk reduction of PsA development may be a pragmatic clinical endpoint in clinical trials, particularly in the subset of patients at high risk for joint disease [62]. It has not yet been established whether the prodromal signs and symptoms of PsA truly represent a prodrome or if they represent manifestations of PsA in patients without an appropriately recognized diagnosis. Therefore, it is difficult to distinguish between the true prevention of PsA development versus the treatment of early, not yet clinically apparent PsA. Prediction algorithms using machine learning might help analyze relevant factors that infer PsA from background events [72]. Further follow-up studies are needed to assess the long-term effects of targeted pharmacotherapies for preventing and/or delaying the development of PsA.

## 6. Targeted Pharmacotherapies May Reduce the Burden of the Cumulative Life Course Impairment in Psoriatic Patients

The burden of psoriasis may be considerable and substantially affect the physical, psychological, and socio-economic aspects of the patient’s life. Therefore, long-term uncontrolled psoriasis may progressively and negatively impact the life of a patient, limiting opportunities, influencing major life-changing decisions (MLCDs), and eventually hindering life potential [74,75].

The cumulative life course impairment (CLCI) represents a concept describing the incremental burden over time of dermatological diseases and the subsequent chronic impairment associated with persisting psychological, social, and personal damage [75,76]. To date, studies on the burden of psoriasis have focused on the quality of life (QoL), measuring the impact of psoriasis at a specific point in the patient’s life. However, these cross-sectional, mostly retrospective, studies are not able to identify the progressive impairment that psoriasis patients accumulate over the course of their lives. CLCI aims to overcome this limitation. However, CLCI is specific to a single patient and is difficult to assess. Thus far, no validated tools to determine CLCI are available, and there is still discussion about which variables should be included.

### 6.1. Social Stigma

Several studies showed that patients with a disease such as psoriasis, which has visible symptoms, might develop feelings of stigmatization and rejection [77,78]. Indeed, psoriasis can significantly affect the patient’s self-image and induce embarrassment [79]. Recent studies showed that ~80–90% of patients reported some degree of psoriasis-related discrimination and stigmatization, which had negative effects on their professional careers and personal lives [80,81]. This is associated with greater disease severity and a longer duration of psoriasis [81,82]. Up to ~20% of patients with psoriasis have been banned from hairdresser saloons, swimming pools, or gyms [83]. Over time, these reiterated episodes may concur to cause poor self-confidence, low social connection, and failure to achieve a full life potential. In fact, in order to prevent uncomfortable situations due to the stigmatization of their skin disease, patients with psoriasis elaborate strategies to avoid social interaction on public occasions [84]. Often, this attitude becomes so invasive that it induces patients to limit not only social exposure but also the chance to improve a professional career [85] or to create and solidify intimate relationships [86]. Feelings of low self-esteem and personal and social withdrawal caused by stigmatization might predispose to the development of psychological disorders [87], and indeed patients with psoriasis face an increased risk of anxiety and depressive symptoms [87,88]. Up to 60% may develop depression [89] and suicidal ideation, which are associated with long disease duration [90,91,92,93]. Psoriatic patients who experience the aforementioned psychological symptoms may also face a higher risk of addictive behaviors, including alcohol abuse, smoking, drug abuse, and food dependency [94,95,96].

### 6.2. Socio-Economic Status

Another important aspect of CLCI is related to the socio-economic consequences of long-lasting psoriasis. Psoriasis may adversely affect daily activities and quality of life and influence a patient’s potential to earn an income and gain full-time employment. A study from the National Psoriasis Foundation database showed a greater risk of low income (less than USD 30,000/year) in patients with severe psoriasis compared to those with mild disease (*p* < 0.001) [96]. Other data suggest a negative effect of psoriasis on professional careers [85]. It has been reported that patients with psoriasis encounter difficulties finding or keeping a job, given lower productivity and loss of working time (days or hours) due to the management of flares [85].

### 6.3. Young Patients

It is reasonable to assume that early development of psoriasis during adolescence and early adulthood, when patients are building their personality, establishing social contacts and partnerships, and planning education and professional careers, will have a greater effect on the life course compared to a later onset of psoriasis. Limitations in the ability to do regular activities, such as studying and exercising, were reported by 59.2% of the patients surveyed [97]. Moreover, at this age, fewer intrapersonal coping mechanisms, which may protect from CLCI, are available. An increased risk for incident suicide ideation among pediatric patients with psoriasis has been found [98]. Therefore, a permanent change in a young patient’s life course may represent a “life scar” with irreversible consequences.

### 6.4. The Potential Role of Targeted Pharmacotherapies

The detrimental impact of psoriasis on the life course could be reduced via psychosocial interventions, such as patient education to improve coping and QoL [99] and/or via an early and effective therapeutic intervention. Randomized controlled trials demonstrated that patients with psoriasis treated with targeted pharmacotherapies showed a significant improvement in QoL, measured with the DLQI score, compared with other treatments [18]. It is reasonable to assume that this improvement in QoL is stronger the more effective the treatment is. Moreover, an improvement in anxiety symptoms with psoriasis therapy has been reported [100] in association with better clinical outcomes [19]. Biologic use is also associated with a lower risk of depressive symptoms when compared with placebo [101] or conventional therapy [20]. Accordingly, it seems plausible that in patients with moderate-to-severe psoriasis, prompt control of the disease may prevent the cumulative impact of psoriasis—physical, psychological, and social—on a patient’s life course.

## 7. Conclusions

In conclusion, chronic plaque psoriasis is an immune-mediated skin disease with a chronic relapsing course, affecting up to nearly 2–3% of the general adult population worldwide. The interleukin (IL)-23/Th17 axis plays a key role in the pathogenesis of this skin disease and may represent a critical target for new pharmacotherapies. The natural history of psoriasis could evolve through progressive phases, including disease progression/extension, development of comorbidities such as PsA and/or intestinal bowel diseases, and the cumulative burden on patients’ psychological and social well-being. Preliminary studies, both in vitro and in vivo, suggest that new targeted pharmacotherapies may exert a *disease modification* effect by normalizing the gene expression and the phenotype of immune cells infiltrating the post-lesional psoriatic skin, including the frequencies of the tissue-resident T memory cells. Moreover, early observational studies suggest that new targeted pharmacotherapies may prevent the transition from psoriasis to PsA. However, these early results need to be confirmed in larger randomized controlled trials. Further studies are also needed to better elucidate whether early treatment with new targeted pharmacotherapies can translate into the prevention of cumulative life course impairment.

## Figures and Tables

**Figure 1 ijms-23-12849-f001:**
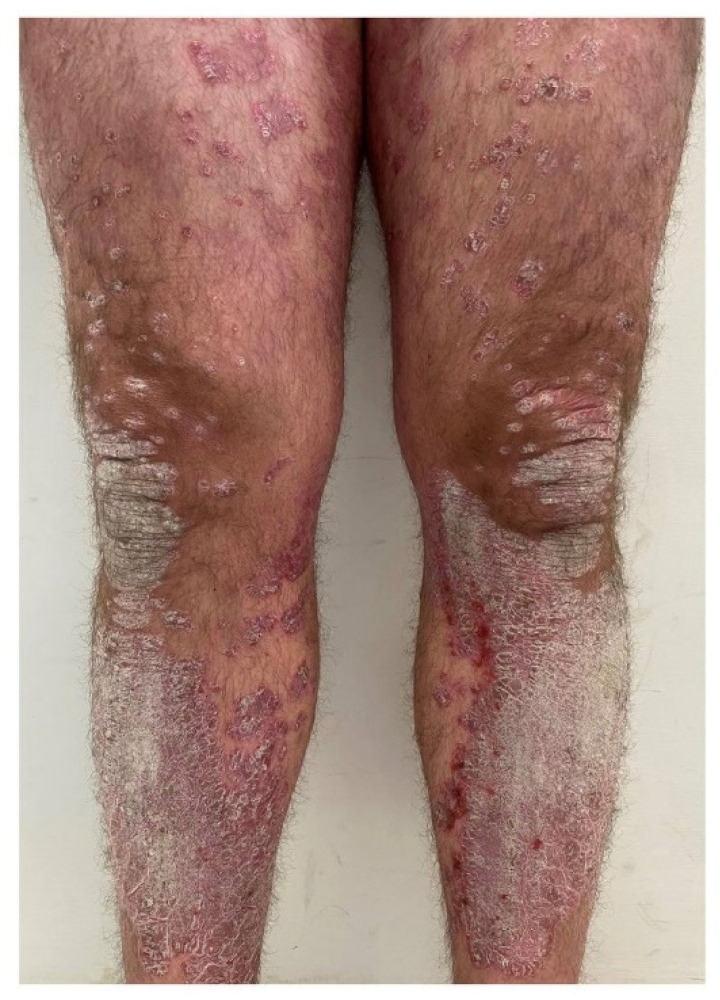
Plaques of psoriasis affecting the extensor surface of lower limbs in a 42-year-old male patient.

**Figure 2 ijms-23-12849-f002:**
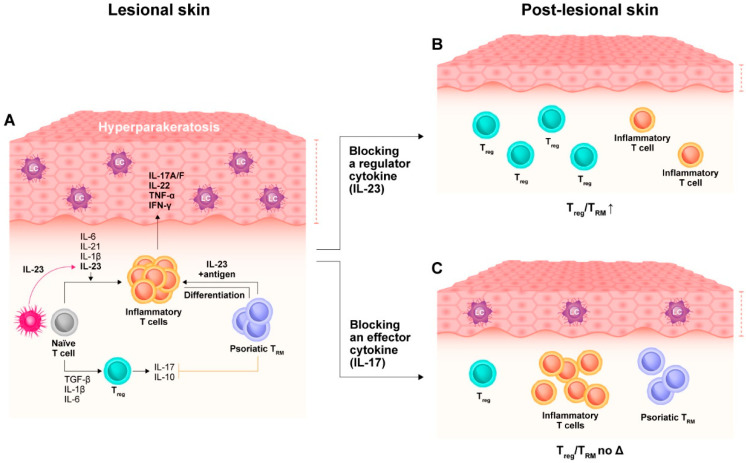
The immunologic scar occurring in post-lesional psoriatic skin. Lesional psoriatic skin (**A**) is infiltrated by inflammatory cells mainly producing interleukin (IL)-23 and IL-17. Dendritic cells (DC) and Langerhans cells (LC) contribute to the differentiation and activation of inflammatory T cells through the production of IL-23. Once activated, *naïve* T cells and resident memory T cells (T_RM_) secrete large amounts of pathogenic cytokines such as IL-17A/F, IL-22, tumor necrosis factor (TNF)-α, and interferon (IFN)-γ acting on inflammatory T cells. Upon successful response to either IL-23 (**B**) or IL-17 (**C**) inhibitors, differential effects on the immune cell compartment characterizing post-lesional psoriatic skin may be observed, with a more marked residual infiltration of LC and T_RM_ that is detected after IL-17 inhibition compared with IL-23 blockade. The IL-23 blockade also preserves the number of T_reg_ cells in post-lesional skin, increasing the T_reg_/T_RM_ ratio, conversely to IL-17 inhibition.

**Figure 3 ijms-23-12849-f003:**
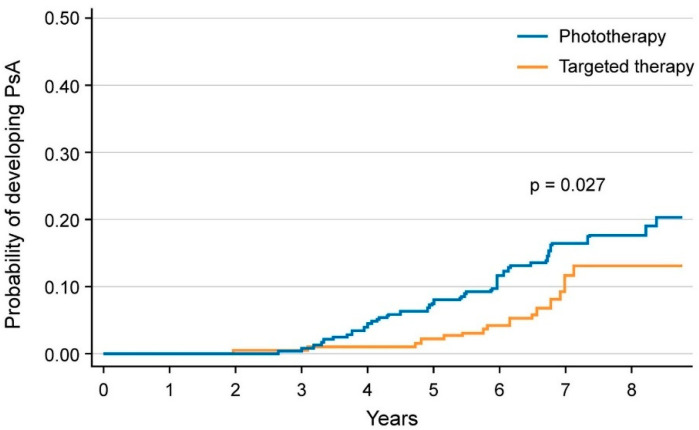
Cumulative incidence rates of psoriatic arthritis in the biological disease-modifying antirheumatic drugs (bDMARDs) versus narrow-band ultraviolet light B phototherapy groups, estimated by the Kaplan–Meier analysis (*p* = 0.027 by the log-rank test). Data derived from Gisondi et al. [14].

**Table 1 ijms-23-12849-t001:** Targeted pharmacotherapies exerting a possible “disease modification effect” in chronic plaque psoriasis.

Area of Influence of Pharmacotherapies	Treatment(s)	Results
Immunophenotype of post -esional skin	Ixekizumab vs. etanercept vs. placebo	Downmodulation of genes involved in multiple inflammatory pathways to a greater extent after ixekizumab compared to etanercept [4]
	Brodalumab vs. etanercept vs. ustekinumab	Suppression of genes involved in multiple inflammatory pathways to a greater extent after brodalumab compared to ustekinumab and etanercept [10]
	Guselkumab vs. secukinumab	Higher Treg/T_RM_ ratio after IL-23 compared to IL-17 blockade [11]
Transition from psoriasis alone to PsA	Ustekinumab	Reduction of subclinical entheseal inflammation scores [12]
	Secukinumab	Improvement of arthralgia, PsAMRIS and synovitis subscore [13]
	TNF-α, IL-17 and IL-12/23 inhibitors vs. NB-UVB phototherapy	Lower risk of incident PsA in patients on biologics compared to phototherapy [14]
	TNF-α, IL-17 and IL-12/23 inhibitors vs. topicals and cDMARDS	Lower risk of incident PsA in patients on biologics compared to topicals but not to cDMARDs [15]
	TNF-α inhibitors, secukinumab and ustekinumab and cDMARDS vs. topicals/no treatment	Lower risk of incident PsA in patients on biologics and cDMARDs compared to topicals/no treatment [16]
	Biologics ^§^ vs. non-biologics	Lower risk of incident PsA in patients on biologics compared to non-biologic treatments [17]
	Biologics ^§^ vs. cDMARDS and phototherapy	Higher risk of incident PsA in patients on biologics compared to cDMARDS and phototherapy
Quality of life	Biologics vs. conventional treatments	Significant reduction of the DLQI score after biologics compared to other treatments [18]
Anxiety	Biologics	Improvement in anxiety symptoms [19]
Depression	Biologics	Lower risk of depressive symptoms [20]

*Abbreviations*: PsAMRIS: psoriatic arthritis magnetic resonance imaging scoring system; PsA: psoriatic arthritis; cDMARDS: conventional disease-modifying antirheumatic drugs; DLQI: dermatology life quality index. ^§^ TNF-α: tumor necrosis factor, interleukin (IL)-17, interleukin (IL)-12/23, interleukin (IL)-23 inhibitors; T reg: T regulatory cells; T_RM:_ tissue-resident memory cells.

## Data Availability

Not applicable.

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
