# Peer review of "Could Targeted Pharmacotherapies Exert a “Disease Modification Effect” in Patients with Chronic Plaque Psoriasis?"

_ijms, 2022, doi:10.3390/ijms232112849_

Round 1

Reviewer 1 Report

I like the flow of the article but would appreciate a table summarizing the transcriptomic data, PsA prevention etc effects of biologics as well as the conventional drugs.

Author Response

Verona, 20st October 2022

Prof. Dr. Maurizio Battino

Editor-in-Chief

Prof. Dr. Annamaria Offidani

Guest Editor

Dear Prof. Dr. Offidani,

We respectfully resubmit our revised manuscript entitled “Could targeted pharmacotherapies exert a “disease modification effect” in patients with chronic plaque psoriasis?”.

We would be grateful if you could consider the revised manuscript for publication in the Special issue “Skin, Autoimmunity and Inflammation” of International Journal of Molecular Sciences.

We wish to thank again the two Reviewers for their constructive comments, which have further improved the quality of the manuscript.

Please find enclosed the revised manuscript and a point-by-point response to the Reviewers’ comments.

Yours sincerely,

Paolo

Prof. Paolo Gisondi

Section of Dermatology and Venereology,

Department of Medicine, University of Verona

Piazzale A. Stefani 1, 37126 Verona, Italy

E-mail: paolo.gisondi@univr.it

ijms-1959833 - Could targeted pharmacotherapies exert a “disease modification effect” in patients with chronic plaque psoriasis?

Point by point response to Reviewer #1

REVIEWER COMMENTS

I like the flow of the article but would appreciate a table summarizing the transcriptomic data, PsA prevention etc effects of biologics as well as the conventional drugs.

AUTHORS REPLY

We thank the Reviewer for the supportive comment. As requested, we included now a table summarizing the findings of the studies focussing on: (1) transcriptomic data; (2) PsA prevention, including the effects of either biologics or conventional drugs; and (3) quality of life and psychological burden. Please, find the table below.

Table 1. Targeted pharmacotherapies exerting a possible “disease modification effect” in chronic plaque psoriasis.

Area of influence of pharmacotherapies

Treatment(s)

Results

Immunophenotype of post lesional skin

Ixekizumab vs etanercept vs placebo

Downmodulation of genes involved in multiple inflammatory pathways to a greater extent after ixekizumab compared to etanercept [4]

Brodalumab vs etanercept vs ustekinumab

Suppression of genes involved in multiple inflammatory pathways to greater extent after brodalumab compared to ustekinumab and etanercept [36]

Guselkumab vs Secukinumab

Higher Treg/TRM ratio after IL-23 compared to IL-17 blockade [40]

Transition from psoriasis alone to PsA

Ustekinumab

Reduction of subclinical entheseal inflammation scores [63]

Secukinumab

Improvement of arthralgia, PsAMRIS and synovitis subscore [64]

TNF-, IL-17 and IL-12/23 inhibitors vs NB-UVB phototherapy

Lower risk of incident PsA in patients on biologics compared to phototherapy [65]

TNF-, IL-17 and IL-12/23 inhibitors vs topicals and cDMARDS

Lower risk of incident PsA in patients on biologics compared to topicals but not to cDMARDs [66]

TNF- inhibitors, secukinumab and ustekinumab and cDMARDS vs topicals/no treatment

Lower risk of incident PsA in patients on biologics and cDMARDs compared to topicals/no treatment [67]

Biologics§ vs non-biologics

Lower risk of incident PsA in patients on biologics compared to non-biologic treatments [68]

Biologics§ vs cDMARDS and phototherapy

Higher risk of incident PsA in patients on biologics compared to cDMARDS and phototherapy

Quality of life

Biologics vs conventional treatments

Significant reduction of the DLQI score after biologics compared to other treatments [97]

Anxiety

Biologics

Improvement in anxiety symptoms [99]

Depression

Biologics

Lower risk of depressive symptoms [101]

Abbreviations: PsAMRIS: psoriatic arthritis magnetic resonance imaging scoring system; PsA: psoriatic arthritis; cDMARDS: conventional disease modifying antirheumatic drugs; DLQI: dermatology life quality index.

  • TNF- Tumor necrosis factor, Interleukin (IL)-17, Interleukin (IL)-12/23, Interleukin (IL)-23 inhibitors; T reg T regulatory cells; TRM Tissue resident memory cells.

ijms-1959833 - Could targeted pharmacotherapies exert a “disease modification effect” in patients with chronic plaque psoriasis?

Point by point response to Reviewer #2

REVIEWER COMMENTS

The article also reviews the state of the art of the immunopathogenesis of psoriasis and how targeted pharmacotherapy could be a "new therapeutic tool", especially in improving the quality of life of this type of patients. This narrative review is well written, pleasant to read, with a clear and defined structure. Its conclusions answer the research question posed.

AUTHORS REPLY

We thank the Reviewer for the supportive comment.

Reviewer 2 Report

The article also reviews the state of the art of the immunopathogenesis of psoriasis and how targeted pharmacotherapy could be a "new therapeutic tool", especially in improving the quality of life of this type of patients. This narrative review is well written, pleasant to read, with a clear and defined structure. Its conclusions answer the research question posed.

Author Response

Verona, 20st October 2022

Prof. Dr. Maurizio Battino

Editor-in-Chief

Prof. Dr. Annamaria Offidani

Guest Editor

Dear Prof. Dr. Offidani,

We respectfully resubmit our revised manuscript entitled “Could targeted pharmacotherapies exert a “disease modification effect” in patients with chronic plaque psoriasis?”.

We would be grateful if you could consider the revised manuscript for publication in the Special issue “Skin, Autoimmunity and Inflammation” of International Journal of Molecular Sciences.

We wish to thank again the two Reviewers for their constructive comments, which have further improved the quality of the manuscript.

Please find enclosed the revised manuscript and a point-by-point response to the Reviewers’ comments.

Yours sincerely,

Paolo

Prof. Paolo Gisondi

Section of Dermatology and Venereology,

Department of Medicine, University of Verona

Piazzale A. Stefani 1, 37126 Verona, Italy

E-mail: paolo.gisondi@univr.it

ijms-1959833 - Could targeted pharmacotherapies exert a “disease modification effect” in patients with chronic plaque psoriasis?

Point by point response to Reviewer #2

REVIEWER COMMENTS

The article also reviews the state of the art of the immunopathogenesis of psoriasis and how targeted pharmacotherapy could be a "new therapeutic tool", especially in improving the quality of life of this type of patients. This narrative review is well written, pleasant to read, with a clear and defined structure. Its conclusions answer the research question posed.

AUTHORS REPLY

We thank the Reviewer for the supportive comment.